# Challenges of Coal Mining Regions and Municipalities in the Face of Energy Transition

**Marek Cała, Anna Szewczyk-Świątek and Anna Ostręga\***

Faculty of Civil Engineering and Resource Management, AGH University of Science and Technology, 30-059 Kraków, Poland; cala@agh.edu.pl (M.C.); aswiatek@agh.edu.pl (A.S.-Ś.)
* Correspondence: ostrega@agh.edu.pl

**Abstract:** The energy transition currently taking place in the mining regions of the European Union poses many challenges that need to be addressed with a view to 2030 and 2050, of which the reduction of greenhouse gas emissions is the key one. Initial results of the research project entitled "Models of a transition to a climate-neutral, circular economy for mining regions under transformation process", which is developed in parallel with the transition of mining town Brzeszcze, are presented. The challenges, in the context of energy transition, for both the EU and local governments were identified on the basis of EU policies and the experience of the project team from the cooperation with the commune of Brzeszcze. A "research by design" method was used to develop model solutions. In the context of local challenges, there was a discussion of the Green Deal objectives and "greenery" as a tool for transformation and achieving well-being. It was concluded that a comparison of the tangible (mining and social infrastructure) and intangible (privileges) well-being provided by a "carbon-based" economy with the new "well-being" weights in favour of the new. This is reflected in the concerns of mine workers and the citizens as well. Therefore, proposing appropriate revitalization of a post-mining site will be one of the challenges. With regard to the revitalization, a discussion was held on the role of mining heritage which can trigger either a "growth machine" or a "decline machine" depending on the decisions taken, compatible or not with a circular economy.

**Keywords:** EU policy; society; climate; just energy transition; challenges; revitalisation; circular economy

## 1. Introduction

Historically, coal has played an important role in the European economy. Today, the coal sector is still present in 12 EU countries and 41 regions. It consists of 128 coal mines (of which 79 are hard coal) with an annual output of approximately 500 million tons (55% of gross EU consumption). By contrast, coal infrastructure is present in 108 regions. An estimated 237,000 people are employed in coal mining, of whom 185,000 are in hard coal mining. More than half of the employees in the coal sector work in Poland [1]. About 10,000 people are employed in peat extraction and 6000 in shale exploitation [2]. A total of 207 coal-fired power plants function in 21 EU countries with a total power capacity almost 150 GW. Coal-fired power plants employ 52,000 people, including 13,000 from Poland. It is assumed that by 2025 there will have been the first wave of decommissioning including the least efficient power plants, resulting in the loss of about 15,000 jobs, and another wave by 2035 will have resulted in the dismissal of another 18,000 people [1]. To the numbers cited above, one must add the workers in the mining-related industries, which will only outline the true scale of the social problem associated with the energy transition. For the estimation of indirect jobs in mining, a ratio of 0.9 to 3.9 is assumed [1]. The question arises whether "green" jobs will compensate for those lost in the mining and

power industry. According to the Energy-Climate Policy Scenario for Poland, implementation of policies and measures for climate and energy targets will result in an increase in employment (from 15,855 thousand in 2020 to 16,060 thousand in 2040, which is 5 thousand more compared to the scenario without implementation of climate and energy targets) [3]. However, these are forecast data. The energy transition will also result in an increased supply of land transformed by mining and power sector activities, together with infrastructure, restoration and revitalisation of which it will require support.

Fossil fuels still account for a significant share of electricity and heat generation (31% in total) and the largest share of greenhouse gas emissions (53%) (Figures 1 and 2).

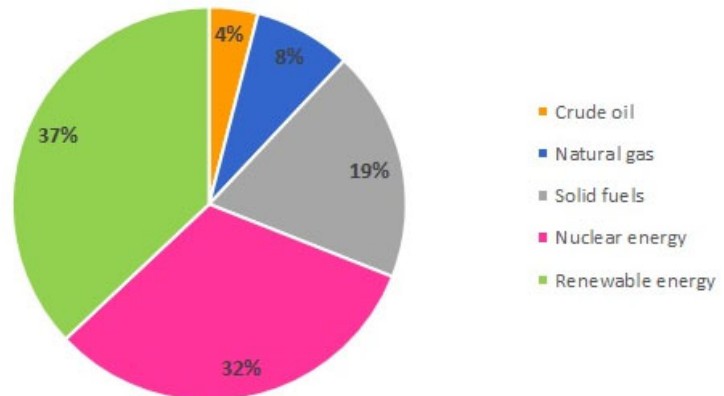

**Figure 1.** Share of EU energy production by source, 2019 (source: [4]).

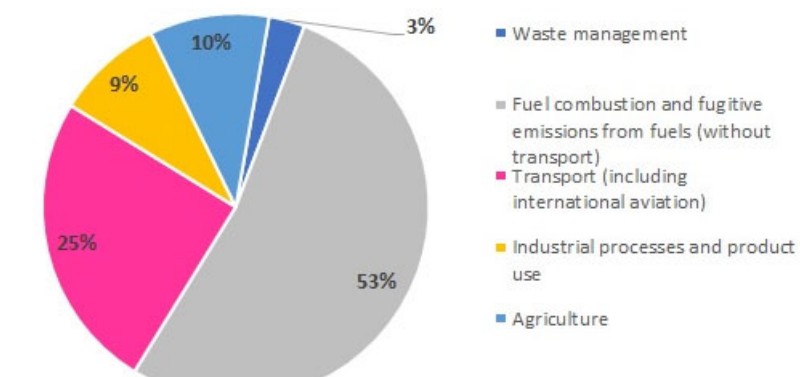

**Figure 2.** Greenhouse gas emissions, analysis by source sector, EU-27, 2018 (source: [4]).

In the case of Poland, although the share of the power industry in $CO_2$ emissions is the highest, it has a decreasing trend, in contrast to emissions from transport (Table 1). In the context of the discussion in this article about preserving or decommissioning mine infrastructure, data on the impact of construction on $CO_2$ emissions are also relevant (adaptation is less energy consuming than building new facilities). Projections beyond 2020 predict successive decreases in $CO_2$ in most sectors, resulting from the implementation of the Energy and Climate Policy for Poland, with a simultaneous increase in GDP [3].

**Table 1.** Projected CO₂ emissions by sector for the Energy and Climate Policy scenario [3].

| Source Category | CO₂ Emission [kt] | | | | | | | |
|---|---|---|---|---|---|---|---|---|
| | **2005** | **2010** | **2015** | **2020** | **2025** | **2030** | **2035** | **2040** |
| Total excluding LU-LUCF | 322,545.79 | 333,457.41 | 312,320.56 | 311,227.40 | 292,568.10 | 268,601.18 | 230,561.04 | 208,893.98 |
| Total including LU-LUCF | 271,331.36 | 298,727.57 | 280,636.39 | 277,532.70 | 263,260.92 | 244,996.85 | 210,897.79 | 193,078.08 |
| 1. Energy | 304,748.07 | 315,601.31 | 292,619.07 | 290,147.24 | 271,155.63 | 246,879.43 | 208,592.08 | 186,661.77 |
| A. Combustion of fuels | 301,576.50 | 312,796.48 | 288,368.88 | 285,598.27 | 266,993.84 | 242,923.81 | 204,975.28 | 183,415.11 |
| 1. Energy industries | 177,290.03 | 172,262.80 | 162,622.03 | 146,578.98 | 142,112.87 | 132,233.28 | 101,830.10 | 87,259.45 |
| 2. Manufacturing and construction | 33,790.32 | 29,455.75 | 27,738.32 | 25,437.57 | 22,234.82 | 19,355.89 | 17,432.54 | 15,639.33 |
| 3. Transport | 35,613.78 | 48,659.65 | 47,367.83 | 62,849.34 | 60,362.78 | 56,327.76 | 54,598.87 | 52,365.71 |
| 4. Other sectors | 54,882.37 | 62,418.29 | 50,640.71 | 50,732.37 | 42,283.37 | 35,006.87 | 31,113.77 | 28,150.62 |
| B. Fugitive emissions from fuels | 3171.57 | 2804.83 | 4250.19 | 4548.97 | 4161.80 | 3955.62 | 3616.79 | 3246.65 |
| 1. Solid fuels | 2019.08 | 1747.97 | 2221.01 | 2521.42 | 2133.60 | 1926.90 | 1587.64 | 1217.12 |
| 2. Crude oil and natural gas | 1152.49 | 1056.85 | 2029.18 | 2027.55 | 2028.20 | 2028.72 | 2029.16 | 2029.53 |
| 2. Industrial processes and product use | 16,091.78 | 16,642.81 | 18,484.19 | 19,327.17 | 19,622.99 | 19,909.94 | 20,129.36 | 20,344.52 |
| 3. Agriculture | 1291.94 | 790.01 | 736.36 | 1013.16 | 1041.93 | 1064.27 | 1092.06 | 1140.15 |
| 4. Land Use, Land Use Change and Forestry (LULUCF) | −51,214.43 | −34,729.84 | −31,684.16 | −33,694.70 | −29,307.18 | −23,604.33 | −19,663.26 | −15,815.90 |
| 5. Waste (incineration and combustion) | 414.00 | 423.27 | 480.95 | 739.83 | 747.54 | 747.54 | 747.54 | 747.54 |
| CO₂ emissions from biomass | 19,803.98 | 30,442.05 | 34,962.70 | 41,228.70 | 42,222.21 | 45,167.75 | 47,522.40 | 50,028.71 |

The threats posed by climate change and environmental degradation determine the formulation of ambitious challenges on a global scale [5–9]:

- reducing greenhouse gas emissions by at least 55% by 2030 (compared to 1990 levels) and achieving net zero by 2050;
- ensuring at least a 32% share of energy from renewable sources in the overall energy consumption;
- improving energy efficiency by at least 32.5%;
- keeping the global temperature increase to well below 2 °C and pursuing efforts to keep it to 1.5 °C;
- developing a competitive, resource-efficient, circular economy;
- protection and restoration of biodiversity;
- achieving energy efficiency and affordability in the construction sector;
- carrying out the transformation in an equitable manner so that no person or region is left behind;
- reusing of brownfield sites with geotechnical or other constraints.

The Oxford Advanced Learner's Dictionary [10] explains "challenge" as a "new or difficult task that tests somebody's ability and skill". In turn, Cambridge Dictionary [11] defines "challenge" as "something that needs great mental or physical effort in order to be done successfully and therefore tests a person's ability". Based on these definitions, in the context of the issues addressed in the following paper, climate challenges remain a task and, at the same time, a test of knowledge and skills of an individual person and the world community alike. In order to meet the global challenges, specific actions need to be taken in the fields of politics and legislation, environment, economy and society, which should not suffer from the transformation. In the context of this article, however, it is most

important to characterise the challenges faced by local communities (here Brzeszcze) as consequences of global political decisions (challenges being ideas about how something that is common should be shaped in the future) and to describe how "big ideas" are translated into selected operational practices.

The subject of the study is the commune of Brzeszcze which has been connected with coal exploitation for 118 years, and in particular the area of the closed Brzeszcze East Hard Coal Mine, which requires reclamation and revitalisation. Revitalisation in the context of energy transition is supposed to put social and economic investments in post-mining areas on a new track, leading towards climate neutrality. In this context, nature-based solutions are valued, but also those that reuse the residues (wastes) of (especially mining) activities. While in revitalisation it is desirable to involve the inhabitants in the process and to let them participate in the decision-making, in transformation it is considered most important to support them in retraining and taking up new jobs.

The example of Brzeszcze commune will be used to illustrate the challenges faced by local governments, mine workers and residents, and how "false consciousness" can thwart the achievement of their desired results. The aim of the case study is to analyse how idea of greenery in "green deal" is understood and combined with local development policies, how it can be tuned to start transformation process (and, by the way, save industrial obsolete buildings). Preliminary results of the work on a research project entitled "Models of a transition to a climate-neutral circular economy for mining regions in transition", carried out at AGH University of Science and Technology in the framework of the Initiative for Excellence – Research University, will be presented.

## 2. From International Policies for a Just Transition to Local Development Policies

The history of the concept of just transition dates back to the 1950s when the Fund for the Retraining and Resettlement of Workers was established within the European Coal and Steel Community. This fund was intended to facilitate retraining for workers who had lost their jobs as a result of new technological developments [12]. Although the term "just transition" was not used at the time, the objectives of this fund were similar to those formulated today (retraining, providing assistance in finding a job and, if other measures fail, financing their relocation to a region with greater employment opportunities). This fund was transformed in the Treaty of Rome (1957) into the European Social Fund (ESF), dedicated to workers in modernised industries such as coal mining [12]. The term "just transition" was introduced into the policy debate in the 1990s by North American trade unions supporting workers who had lost their jobs due to tightening environmental policies [12].

Subsequent initiatives concerned climate care, with the long-term consequence of phasing out mines and power stations based on fossil fuels. The United Nations Framework Convention on Climate Change (UNFCCC), which was one of the outcomes of the Rio Earth Summit (1992), set the goal of "bringing about stabilisation of greenhouse gas concentrations in the atmosphere at a level that would prevent dangerous anthropogenic interference with the climate system". The Convention set out principles, commitments and ways to achieve them (through policies, programmes, research, technology, education and awareness raising). It emphasized reliance on principles of equity, understood as the distribution of efforts between more and less developed countries [13].

The COP21 in Paris (2015) adopted the Climate Agreement, recognised as a universal and legally binding act (ratified by 190 countries, including the European Union). The agreement sets out a global action plan to prevent global warming and improve countries' ability to cope with the impacts of climate change [14]. The implementation of these commitments is to be enabled by the detailed principles contained in the so-called Katowice Climate Package (adopted at COP24 in Poland (2018) [15]. It was raised at that time that the transition must take place in a fair and solidarity-based manner.

Published in 2015 by the International Labour Organisation, the "Guidelines for a Just Transition" provide a set of principles for the transformation of economies and societies to an environmentally sustainable model. Among the principles, in addition to establishing a comprehensive policy framework, there is the need for a strong social consensus on sustainable development goals and on how to achieve them, as well as the need for social dialogue at all stages and levels of governance [16].

To ensure a socially just coal phase-out, in December 2017 the European Commission established the Platform on Coal Regions in Transition. The said Platform aims to support countries, regions, communities and workers to meet the challenges of the transition to clean energy and related economic diversification [17].

A communication on the European Green Deal was issued in 2019 [18]. Addressing climate and natural environmental issues was identified as the most important task facing the current generation. The idea of the Green Deal is reflected in the diagram (Figure 3).

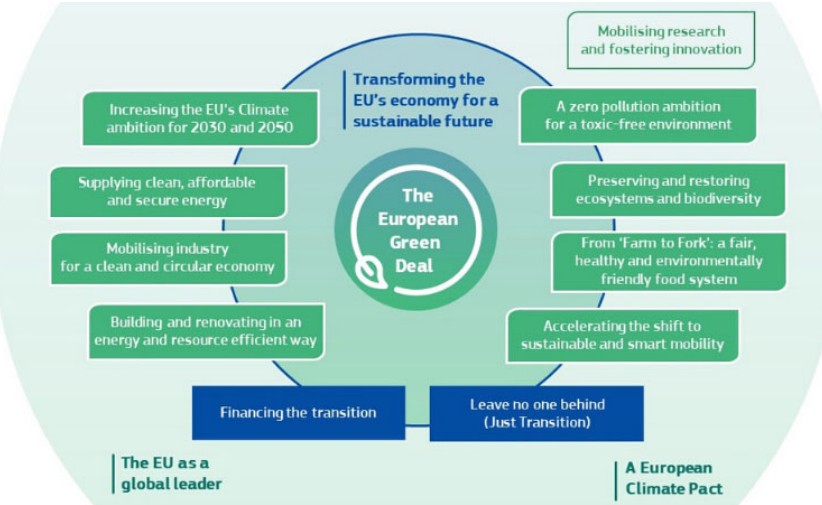

**Figure 3.** The concept of the European Green Deal (source: [18]).

It is worth noting that (broadly defined) environmental objectives dominate over social objectives in the Green Deal. The rhetoric of widespread acceptance of the Green Deal is built around: lowering emissions, providing clean, affordable and safe energy, reducing waste generation and reusing waste (with a particular emphasis on construction), reducing pollution generated by transport, ensuring high quality food, protecting and restoring ecosystems and biodiversity [18]. In all likelihood, it can be said that the difficult consequences of global challenges faced by local communities are more easily recognised as necessary by the public when the overriding objective is to protect the planet. When the battle for the future of the planet is fought, it is easier to accept that mining towns will lose their economic basis. The European Green Deal aims to *protect the health and well-being of citizens against environmental risks and negative impacts* [2], but in the aspects of social well-being and material infrastructure this sustaining well-being offers citizens much less than the carbon economy (It is worth mentioning in this context that the carbon economy based on large, state-owned enterprises (as opposed to a low-carbon economy targeting mainly small and medium-sized, private companies), which offered significantly more social benefits and amenities—both soft (e.g., allowances, subsidies, vouchers) and hard, present-day in spaces, e.g., factory parks, sports facilities, community centres, common rooms, libraries, kindergartens, schools, holiday centres, etc. In this context, it is worth emphasising that the Green Deal downplays the importance of the material elements of planning—urban planning, architecture (which is particularly significant in relation to post-industrial and "empty" areas). Thus, it reinforces the problem (raised by Koolhaas almost 30 years ago [19]) of downplaying the importance of urban planning, while urbanising the

territory on an unprecedented scale [18]). "Well-being" of the Green Deal compared to "well-being" expressed materially (also in the built space of the "old order"), is a much more ephemeral concept.

It would be impossible to cite all the documents and initiatives related to fair transition and climate protection, but one can venture to say that currently every strategy takes climate objectives into account. The restructuring of the mining sector in the 1990s, for example, showed that retraining the mining workforce was the biggest challenge and one of the reasons for this was strongly rooted traditions. While the first transformation initiatives were aimed at supporting workers, now support will be directed also towards environmental measures and the development of environmentally friendly technologies.

The Just Transition Fund (JTF) is to be the mechanism for implementing the Green Deal and mitigating its social consequences, which introduces *the new growth policy—based on ambitious climate and environmental objectives and participatory processes bringing citizens, cities and regions together in the fight against climate change and for environmental protection* [2]. The document establishing the JTF identifies as its main objective: *enabling regions and people to address the social, economic and environmental impacts of the transition towards a climate-neutral economy*. It is worth noting that the phrases "enabling" and "together in the fighting" prove that the legislator accepts that the work and effort of citizens are necessary and, at the same time, shifts the responsibility for implementing solutions from the central level (the legislator) to the regional and local level (whereas the carbon economy was based on central support). The questions remain open as to whether smaller centres are able to undertake and manage effectively these key transformations, whether such regulation does not add to the burden on those who are in a difficult situation and whether functioning in the reality of competing regions (in the discussed case, particularly important due to the location of Brzeszcze, administratively in the Małopolska (Lesser Poland) region, although economically connected to Silesia) will not pose a threat to the coherent transition.

Activities eligible for support include [2]:

- clean energy and energy efficiency technologies;
- regeneration, decontamination, renaturalization and land redevelopment;
- strengthening the circular economy;
- diversification and creation of new enterprises, including start-ups;
- support for employees (up-skilling and re-skilling, job search assistance, active inclusion);
- digitisation and digital communications;
- research, innovation, technology transfer;

which, in addition to the problems signalled earlier, draws attention to:

- the threat posed by support for (probably mainly) private economic initiatives (which may also result in the privatisation of the environment and, for example, the "privatisation of green energy" exacerbating social inequalities);
- the lack of planning for any material investment of a strictly public nature, aimed at improving social well-being (and reducing the risk of private investment in problematic areas, which at the initial stage of transition does not guarantee a return on the capital invested) or at ensuring energy security
- for citizens (which was provided by coal and, for example, the allowances mentioned previously).

Due to the fact that, according to JTF (as previously mentioned), responsibility for implementing of Green Deal is shifting to the regional and local level of governance, it is necessary to characterize briefly two theories of local development policies that are used in the specific context of transformation of post-industrial areas. The first of them, "growth machine" [20] describes a concept of local development based on cooperation between key investors in real estate and municipal authorities (offering, e.g., subsidies, discounts, write-offs, legislative facilitations), which is characterised as a rational path enabling to

attract large investments to declining cities, accelerating market revitalisation processes (by creating, e.g., enterprise zones, innovation districts). The concept of a "decline machine" (e.g., [21]) has been created by analogy with the "growth machine", which describes a focus on activities that enable value extraction (capital accumulation) from abandoned and unused resources.

In this context, analyses of projects, plans and operational strategies, in terms of the type and degree of realisation for the declared "well-being" and awareness of the challenges they pose to local communities, can be considered of particular research relevance. The following case study is an attempt to do this.

## 3. Materials and Methods

### 3.1. Case Study Analysis

The disused Hard Coal Mine Brzeszcze East (KWK Brzeszcze Wschód) is situated in the western part of the Małopolska (Lesser Poland) region, in the county (powiat) of Oświęcim, commune of Brzeszcze. From the geological-geographical point of view, Western Małopolska is a part of the Upper Silesian Coal Basin, one of the three hard coal basins in Poland, covering about 80% of deposits in the country.

Currently 6 (and previously 7) communes of the Western Małopolska are covered by hard coal mining, with only two mining plants located in Małopolska (Brzeszcze and Janina) (Figures 4 and 5). Taking into account coal processing as well as exploitation and processing of other minerals (e.g., zinc and lead ores, rock raw materials) it should be concluded that the economy of the sub-region is strongly dependent on the high-emission industry. Małopolska is one of the six coal regions in transition in Poland.

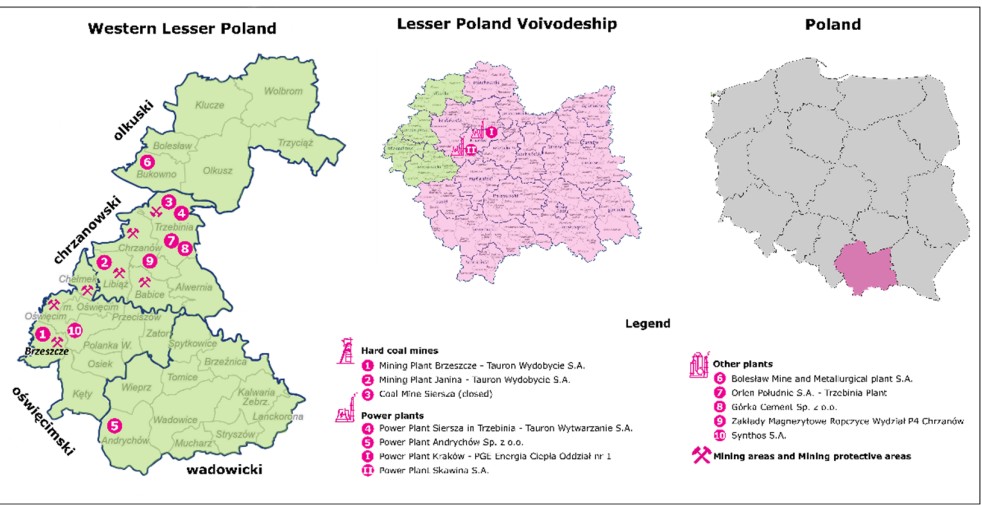

**Figure 4.** Location of mining plants, power plants and other energy-intensive and carbon-intensive enterprises in Western Małopolska (source: own study).

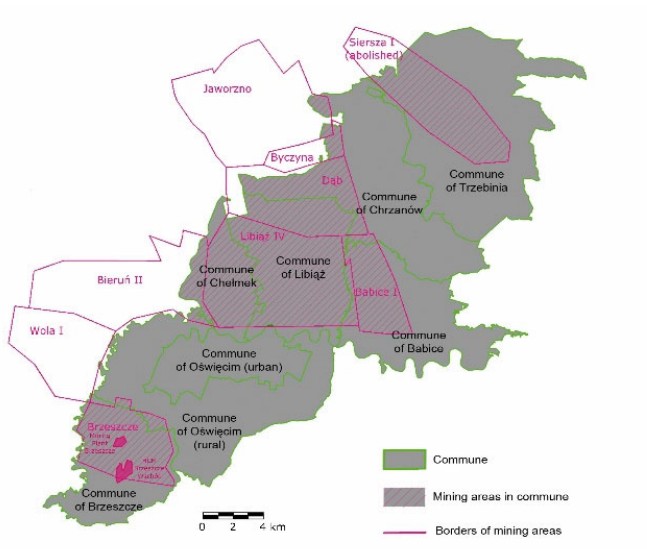

**Figure 5.** Location of mining areas within the Małopolska communes (source: own study based on [22]).

On the basis of an analysis of hard coal resources, the contents of the Energy Policy for Poland until 2040 and the effects of the restructuring of the mining industry to date, a further reduction in the use of coal is assumed, but its permanently significant position in the energy mix is emphasised [23]. In this context it should not come as a surprise that the lack of specific plans for further restructuring of mines, decision on granting the Just Transition Fund for Western Małopolska or information on investments in renewable energy sources, is accompanied by anxiety (and even opposition to changes), mainly among employees of mines and mining companies.

Coal mining in Brzeszcze has been carried out since 1903 [24]. Currently, it is conducted by the Brzeszcze Mining Plant – TAURON Extraction JSC (Zakład Górniczy Brzeszcze – TAURON Wydobycie S.A.). In 1918, in Jawiszowice village, the construction of Jowisz shaft, later called Andrzej III, was commenced. The shaft was later expanded into Branch of Hard Coal Mine Brzeszcze East (KWK Brzeszcze Wschód). The Hard Coal Mine Brzeszcze East was the first and only state-owned hard coal mine constructed during the Second Polish Republic. As an object of particular interest for the state authorities in the years 1927–1937, it underwent major investments, owing to which it possessed the most modern solutions in the mining industry [25]. It was a place of work not only for generations of Brzeszcze citizens, but also for prisoners of Jawischowitz camp (Auschwitz sub-camp) during the Second World War [26].

Thanks to the construction of the mine, the once agricultural settlement became a town (municipal rights in 1962) with over 11,000 inhabitants today. The mine has invested in the erection of educational, sports and recreational, commercial, healthcare and cultural facilities, while numerous social organisations used to operate or are still operating at the mine [24]. Some spectacular, but undoubtedly related to the mine, social motifs include the sport successes of the "Górnik Brzeszcze" Sports Club in the national and international arena, which was a source of pride [27].

The Brzeszcze Mining Plant is still in operation and the Hard Coal Mine Brzeszcze East Branch was decommissioned in 1995. Until 2016 it was used as a material shaft. Part of the area together with the buildings was sold to private investors and redeveloped without a concept, while the remaining area together with the building complexes of the Andrzej III and Andrzej IV Shafts was transferred to the Mine Restructuring Company JSC (MRC, Spółka Restrukturyzacji Kopalń S.A.) in 2016 for liquidation (Figure 6).

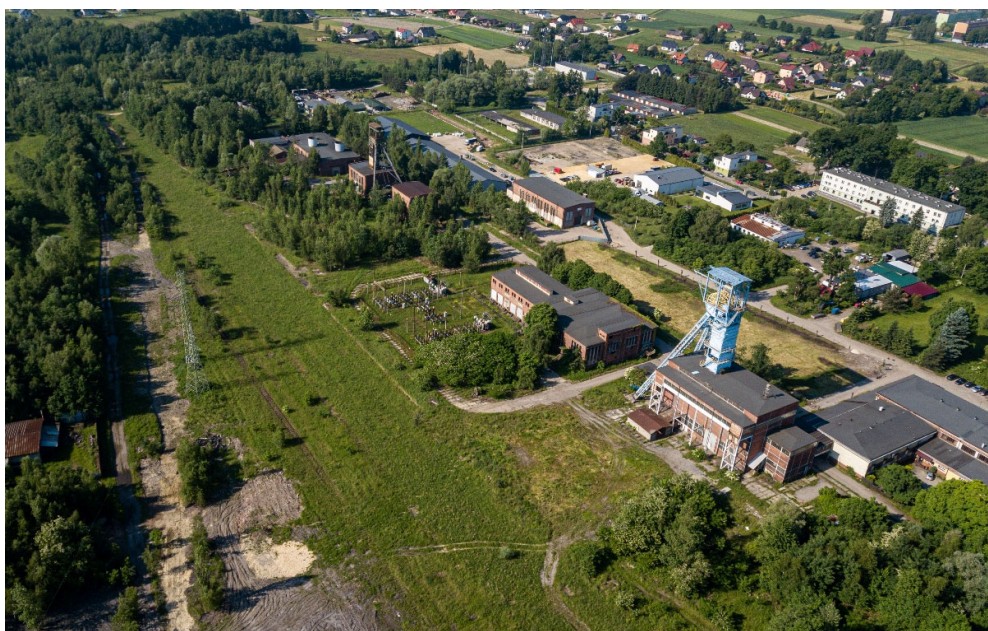

**Figure 6.** Inactive Hard Coal Mine Brzeszcze East, June 2021 (M. Kramarczyk).

Currently it is the object of interest for the Brzeszcze Commune. The complex of buildings including the Andrzej III Shaft in 2011 was entered into voivodship monument records as a complex of modernist industrial buildings. The entry was intended to protect it against demolition, lawlessness, unauthorised reconstruction, but it did not interfere with necessary modernisation and revitalisation of the historical complex [28]. Despite this, some of the buildings were demolished. Many other objects and elements of equipment as well as line infrastructure were liquidated. The post-mining area is adjacent to an extractive industry waste facility and landfill site for municipal waste.

Hence, one of the challenges ahead of the Brzeszcze Commune is an optimal revitalisation of the post-mining area and industrial infrastructure, in view of problems such as:

- the established practice of demolishing the infrastructure and only afterwards looking for ideas for land use or for investors to whom next parts of the post-mining area are granted;
- the different attitudes of decision-makers to the preservation, protection and adaptation of the mining heritage, stemming from a fear of the responsibility to maintain the sites and ensure their safety until adapted to new functions;
- the lack of analyses concerning the real demand for specific job places or economic zones against the background of existing or planned zones.

It is expected that Brzeszcze, like other communes, should join in the implementation of the challenges outlined at the EU level, i.e., striving for climate neutrality, which in the case of Brzeszcze is more difficult because it raises fears of losing privileges related to work in the mine, and easier because, as an area based on a high-carbon economy, it has chances for greater financial support for these transformations.

Taking up such defined challenges, the commune authorities decided to develop a plan for the transition of Brzeszcze, one of the most important elements of which is the revitalisation of the disused Hard Coal Mine Brzeszcze East. For this reason, Brzeszcze became the subject of research conducted at the AGH University of Science and Technology.

*3.2. Methods*

The characterisation concerning the main directions of research related to the problems of transformation of post-mining areas was a prelude to describing the challenges and facing the problem of searching for alternative implementation solutions in a specific location. The transformation plan for Brzeszcze is implemented in the "research by design" formula. Working implementation proposals (inspired by the research into the history of the concept of just transition over the decades) applied to a specific location are intended to identify alternative, innovative solutions that can be generalised in the next stage. An important element of the research were interviews with decision-makers, employees of mining and restructuring companies, residents, representatives of non-governmental organisations who were allowed in the interviews to raise issues they considered important themselves. Comparison of the issues raised in the interviews made it possible to define preliminary guidelines for the revitalisation project, which were confronted with the expectations of the residents (as expressed in the vote of the commune council). The purpose of this confrontation was twofold—it was hoped to gain a better understanding of the needs demonstrated by the local community, but also to promote the adopted assumptions.

The project team simultaneously conduct research and participate in the works related to the preparation of the area of the closed Hard Coal Mine Brzeszcze East for revitalisation, which gives the possibility of almost immediate verification of the proposed solutions. The team advise on the protection of the historical infrastructure of the mine, the advisability of its acquisition by the commune in the form of a donation; carry out environmental, geotechnical and technical research; participate in the preparation of planning documents; develop the concept of revitalisation; support the commune in obtaining funds for investment; and help define ways of involving residents in the process of transition and application of the principles of a circular economy. The initial stage of the project has already revealed new challenges facing both the project team and the local community.

## 4. Results and Discussion

The transformation of mining regions within the framework of a circular economy introduces new principles of space management which may even be regarded as opposite to the traditional ones. Exploited and decommissioned areas are understood as a resource that can be economically used (e.g., redeveloped and supported), as opposed to areas previously considered attractive for investment (the development of which is not supported by the new Green Deal).

A concept, in parallel and widely discussed in the context of the circular economy, is the transformation of cities and regions towards their adaptation to climate change, by organising systems (ecological and social) that minimise the impact of urbanisation over the natural environment. It is also noted that "eco-cities", often created as implementations of experimental technological solutions for adaptation to the phenomena of global warming, are also places where, under the guise of the need for "green growth", social inequalities are often deepened [29]. As a remedy for this state of affairs, support is most often indicated for:

- individual eco-enterprises (or local ones that can be implemented by a centre in transition);
- sustainable lifestyles, i.e., developing (broadly defined) mechanisms for resilience to external crises and dwindling resources (e.g., through green-blue infrastructure projects or, on the other hand, supporting the acquisition of "green" professional qualifications).

However, the first interviews in Brzeszcze indicated, as it had been assumed beforehand, that the residents are not confident that the Green Deal reforms will solve their

problems (mainly related to the expected loss of employment). Being unaware of the concrete plans for the closure of the mines in Brzeszcze and its surroundings, it is difficult for them to plan their further professional future. Moreover, the local community lives in the belief that the mine has "always been there" and will be there, and this belief is reinforced by the attempted mine closure (2015), thwarted by strong protests from miners and local residents. It was also found that there is a lack of a sufficiently strong support of local enterprises which could affect a change in the employment rate on significant scale, no local entrepreneurial leaders interested in the economic transformation of the centre were identified, nor even external investors on whom to base a transformation plan could be possible. Moreover, it was assessed that the social security provided by the presence of a (still) functioning mine (Ruch I Branch, a key employer and tax payer) would be very difficult to achieve under conditions of "green growth" (if at all possible).

In taking on the challenge of developing a transformation plan, the project team felt at an early stage of work it was important to find the answers to the following questions:
Why has "green" become such an important element of the transformation plan?
What Green Deal principles can help build "resilience" to the crisis, in the reality indicating a high probability of the Brzeszcze mine closure scenario coming true?
Is it possible to take action to make the vision of living in a greener environment coincide with the closing of the mine?

### 4.1. The Importance of "Greenery" in Transition

The model of urbanization (construction of physical structures sustaining the possibilities of realization of urban functions), based on the leading importance of natural elements, is closely related to the transformations of the city structure resulting from the changes in industrial production processes. Landscape urbanism (developed since the early 1990s), sets as a particularly important goal the use of undeveloped spaces, the hybridization of urban and non-urban typologies, combining the problems of ecology (previously considered in the context of the non-urban environment) with the structure and form of cities [30]. The popularisation of the "discovery" that previously built-up, exploited and urban areas remain the reservoirs of species of natural value is attributed to urban ecologists (particularly active in the study of urban ecosystems as a whole since the 1970s). The recognition that the occurrence of specific species of flora and fauna is a reflection of the history of economic and cultural changes and land use practices [31] has made it possible to emphasise the importance of natural environmental elements as part of heritage. The preservation of ruderal green areas, in the context of the aforementioned studies, can be seen as an alternative to the preservation of buildings, a measure to protect both the ecological and cultural landscape. However, as stated at the beginning, the main aim of the following article is not to prove the rightness of preserving the identity of a place, its "nature" and culture. In the context of the taken up challenge—planning the transformation of a city based on the coal industry (being aware that planning is not to define an implementation project, but an operational plan to guide rather than determine the urbanisation of an area, implementation of which may be a dynamic process requiring monitoring and modification)—it seems important to consider what function greenery plays in contemporary revitalisation projects and how it was perceived by interlocutors at the preliminary stage of Brzeszcze transformation. Interest in ruderal greenery and spontaneous natural succession persists, both in scientific and political circles, one piece of evidence of which is the support for circular economy (CE) programmes, strategies and initiatives. On the one hand, the lack of necessity to interfere in this process, its "naturalness" makes nature (to a certain degree required by the manager) an unmanned tool for the revitalization of rarely used areas and paradoxically, by the absence of people, more effective (for example, the M2W (Military to Wildlife, [32]) revitalisation model is based on such a mechanism of action). On the other hand it helps to change the image of "moonscapes" into "living" ones. Apart from ecological and cultural functions, the symbolic function of "green" areas cannot be omitted as authentic, wild and "naturally" opposite

to the image of the "moonscapes", popularly associated with industrial activity. It should be noted here, the obvious fact that the "green appearance" does not at all determine its ecological, cultural value—but, on the other hand, its actual (ecological) condition does not determine the value of "greenery" as a symbol. Also in Brzeszcze, a vast post-mining area, closed to the public and unused as a result of natural succession, has become an enclave being an important resource of "green" space for inhabitants (who often admit that post-industrial green areas are one of the strengths of their town).

In addition to assigning a symbolic function, an important step, historically, in the emancipation of green and open spaces (also previously exploited) in transformation projects is to emphasise their importance for the resilience of cities to critical situations—not only as ecological and cultural spaces, but enabling adaptation to climate change—also in economic terms, linking their development to the possibilities of shaping energy independence and reducing energy demand (as a result of lower consumption, e.g., by influencing temperature reduction) and this rhetoric is quite well known and accepted by local government officials. However, what is worth emphasising in this case study, the economic rationale makes it possible to argue both for the need to maintain and manage "green" areas (including urban areas) on a spatially significant scale, with the intention of industrial production (nota bene specified by colour) of "green" energy (e.g., "biofarms") as well as their protection and exclusion from use – methods of management traditionally regarded as contradicting. Moreover, EU Biodiversity Strategy [9] was published as a Communication from the Economic and Social Committee (and the Committee of the Regions) and the need to preserve biodiversity is justified by economic considerations (e.g., the fact that over half of the global GDP depends on nature). Significantly, the document does not discuss, e.g., the deepening of inequalities, the social consequences of (postulated) extension of natural protection (including strict protection) to larger and larger areas—it does not discuss the consequences which, given the planned scale of protection, will affect used spaces (probably both inhabited and economically used), the necessary restrictions on property rights in order to implement the postulated afforestation, urban greening, etc.).

The reuse of brownfield sites within urban areas, conceived as empty spaces ready to be redeveloped for this purpose, however, carries risks: on the one hand, the exclusion of these spaces from social life (in the case of industrial "green space"), on the other hand, their "museification" and exclusion from economic use (in the case of park space). Both approaches are particularly important for the centres created around industry (such as Brzeszcze, erected around a mine), where post-industrial areas, although often considered "empty" and "dead" are built up with buildings maintaining identity. The description of brownfields as empty ("there is nothing there"—recurring in the interviews), inconsistent with the facts (and even with awareness and attachment to the place) is well known in the literature. Research indicates that "dead" sites accommodate more activity than officially recognised (e.g., [33]), it is argued that they should be seen as important elements of contemporary culture (e.g., [34]), and it is also noted that "discarded" spaces allow for greater freedom of investment, which can lead to more innovative redevelopment (e.g., [35]). "Museification" of an area is the main threat to Brzeszcze inhabitants—transforming obsolete industrial buildings into culture-led development is—in their opinion—not profitable for them, and was negatively connoted.

### 4.2. Different Aspects of "Greenery" in Green Transformation

However, even the "green" areas, predestined to receive "green" investments (under the Green Deal), do not enjoy a lot of investment from investors—especially the kind that the residents expect—offering jobs and not having a harmful impact on the environment. In the case of developing the plan for Brzeszcze, the project team was confronted with the problem of the negligible influence of local authorities and residents on obtaining such desired investments—the lack of the final shape for the (governmental or voivodship) rules regarding support programmes is a big challenge for the decision makers. Analysing

the economic environment, it was concluded that there is a lack of potential investors who are interested in investing in the area. There is also no indication that the historic value of the objects and the values of the natural environment (appreciated by former mine workers and people familiar with the area) can significantly improve this situation.

In this socially and economically difficult context, with the uncertainty of public subsidies (the described works related to the preparation of the area of the closed Brzeszcze East Hard Coal Mine for transformation are carried out in the so-called transition period in the context of the programming of EU funds, including the Just Transition Fund), it is difficult to reverse economic trends and set the "growth machine" in motion. This method of local development policy was assessed as effective and desirable to be applied in Brzeszcze—both by the inhabitants and the local authorities. In the interviews, the belief that the obstacle to constructing a "growth machine" is the existence of disused post-industrial buildings in potential investment areas. It was argued that traces of industrial activity, buildings of uncertain technical condition, probably lower the value of land. Although emotional attachment to post-mining buildings was declared, the need for their demolition was indicated as rational and preferable. While the demolition of decaying post-industrial buildings and subjecting the land to natural succession (making it greener because free of development), or preparing the land for investment in this transformed area could fit in with the principles of a circular economy, in the light of further analysis of local conditions it appears to be the opposite solution to that expected by the inhabitants—more likely to shape a "decline machine" than growth. As the processes of urban decline and rebuilding are dynamic and constantly occurring (as e.g., Murray [21] argues), it would be a mistake to see urban ruining as an emergency to be fixed by demolition—but as an equivalent process of profit-making (to speculative profiteering from increased property values) stretching from the scrap metal trade to the reduction of property values and opening the way to the sanctioning of any (even negatively impacting) developments. The group potentially benefiting from the ruining is not small: "*Decline machine*" *bring together city builders ranging from real estate capitalists, property owners, land speculators, landlords, and municipal authorities to stabilize decline by cutting off support and discouraging capital investment in the most devastated neighbourhoods and derelict zones of the city* [21]. Notably, in the spectrum of measures to implement this mechanism, (...) "*Inaction*" [should be considered] *as powerful a weapon in the* "*decline machine*" *arsenal as deliberate action to stem the tide of decline* [21].

Demolishing the buildings of the Hard Coal Mine Brzeszcze East, which are currently managed by the Mine Restructuring Company JSC, in the light of the above considerations is a rational action serving to capitalise the profit from the collapse. At the same time, understanding the mechanism of how the "decline machine" operates, allows with a high probability to state that the profits would be transferred to currently supported "growth machines" located in another place—causing a further reduction in prices of local properties (and, in further consequence, of the whole town). By contributing to the deepening of the critical situation, depriving the inhabitants and the local self-government of their own powers the consent for demolition, although in a way compliant (in a way, because the demolition materials would probably be reused e.g., scrap metal) with the principles of a circular economy, contradicts the idea of transformation "leaving no one behind".

### 4.3. Planning of Deliberate Actions

After a brief analysis—emphasised in interviews and declarations by the site manager—of the plans to capitalise on profits from demolitions, the project team decided on the need for substantive support for local government and residents to immediately counteract the start of the "decline machine". It was agreed that the only chance for an effective intervention—ahead of the demolition of the existing buildings—was for the Commune to take over the site in question.

However, taking over (even free of charge) a significant, partly built-up area by the local government is connected with objective economic difficulties and the necessity to

calculate political consequences of the decisions made. The most frequently raised negative consequences of such a move for the community (already burdened with the threat of closing down the activity of the mine) are the loss of income from property taxes and the necessity to ensure security and protection of the area. It should also be noted that the lack of expert opinions on the technical condition of the buildings caused justified fears among the commune councillors regarding the costs of possible demolition works, should they prove necessary in the future. As far as the expert opinions could not be obtained (due to the necessity of quick decisions and costs), investor's cost estimates for the planned demolition works (prepared on the request of MRC) were analysed. Despite the initial conviction of the residents' representatives about the low utility value of the buildings and their poor technical condition, the site visit and the first assessment of the buildings' condition indicated that these structures, erected just after World War I and World War II, appear to be solidly built structures, especially in comparison with the contemporary ones.

However, what the project team found to be the greatest local resource was not the existence of the buildings themselves or the vastness of the site, but the enormous amount of post-industrial detail (the perceptually engaging furnishings—equipment, tools, etc., of the buildings) and the informal—scenographic atmosphere, eschewing "museification". While Agamben [36] defines *museification* as the *exhibition of the impossibility of use* of things, places (and even people), the buildings of the Brzeszcze East Hard Coal Mine are overflowing with items and objects, often with their original purpose unknown to the "unknowledgeable", whose technical condition and wear gives the impression that they can be freely reused. To see in this "waste" (buildings and their furnishings) the possibility of giving them new functions and meanings is a further argument supporting the fact that there are sensible alternatives to the transient "highest and best use of real estate" ("highest and best use", cf. Murray [21]) which often determines the assessment of the market value.

Obviously, the use of equipment or derelict infrastructure is not an action that solves technical, architectural or urban planning problems, and it does not relieve from the responsibility for larger-scale actions. However, the failure to emphasise their importance in initiating the transformation process may indicate a (wrong) underestimation of their significance, at least at the initial stage of transformation. It probably also reduces the burden of political decisions by appealing to the possibility of quick reuse and affective judgements rather than (as in Brzeszcze) originally verbalised rational ones. The evidence can be the decision of the Municipal Council in Brzeszcze (which, after presenting the possibility of quick adaptation of the buildings) changed its verdict and agreed to take over the problematic area. One can say that in this case the Commune decided to take the position of an active player in the "machine of decline", betting on profiting from the ruins (or, according to the nomenclature of the Green Deal, from "waste"). An attempt was made to capitalise the profits, for the purpose of building a future transformation project—importantly, the capital planned to be raised here is not only economic capital, but also social capital. Generating interest in the area, encouraging people to visit it is to be executed with the hope that following the people, the area will attract profitable and desirable local investment. Completing the list of stakeholders in constructing a "decline machine"– the elite of real estate investors with significant means of economic impact, with users interested in the aesthetics of decline (e.g., tourists interested in industrial archaeology and heritage, but also entrepreneurs from the "creative" and "smart" sectors), without necessarily having ownership rights to the site, expands the applicability of this mechanism in planning.

### 4.4. The Value of Greenery Replacing Buildings in the Transformation

For years, mining families have been living in a polluted environment, which the Green Deal reforms are expected to improve. Nevertheless, it has been diagnosed from the beginning that preserving jobs is higher in the hierarchy of the inhabitants' needs than

improving the environment (here mainly the air). Conclusion is comparable to the results of a pilot study (in connection with smog) on the local community's perception of coal mines in Silesia (276 respondents). The hypothesis there was that the image of mining and coal as a fuel for the local community is negative, and that the mines are not socially responsible enterprises. However, survey results indicated a positive image. According to the respondents, coal mining fundamentally affects economic, social and energy security, and despite the negative impact on the environment and the need for restructuring, mines should not be closed [37]. This observation is not intended to discredit miners as environmentally irresponsible, even though they are often portrayed as such ones in public discourse. Further analysis rather indicates that each stakeholder group opts (more or less overtly) for the possibility of economic gains from post-mining space—they differ in their arsenal of means and justifications that reflect a particular ideology. In particular, as mentioned in the introduction, we are concerned with analysing the challenges that these communities and authorities face and whether their identification influences the willingness to make an attempt. It was considered helpful to answer the question: who does the "green transformation", characterised as it is, serve for if the inhabitants of the mining town are not convinced that it will improve their livelihood? The emerging consequences of the reforms and their possible value for the local community (apart from the mere improvement in the quality of the environment, without questioning its objective value) are the main issue which all further analyses, concepts and strategies should comprehensively take into account. Why is the emphasis put on retraining miners and providing them with jobs, which, as many transformation projects have shown, is rather doomed to failure? Perhaps the issue should be approached a little differently and instead of implementing solutions, communities ought to be helped to identify the challenges they will be able to take up?

Although the miners are reluctant to change (moving away from coal), there is a growing conviction among them that change is inevitable. The attitude of the inhabitants of Brzeszcze to the "green environment" is equally nuanced as to the mining legacy (which they appreciate but propose to demolish). The economic importance of natural areas, their influence on increasing the land rent and improving the quality of life of the inhabitants are well known in urban planning theory and practice (Going back to the early days of urban industrialisation and projects such as the revitalisation of a quarry and its transformation into the Buttes Chaumont Park (1864–1867, Paris) by Haussmann and Alphand in order to speculate (not always successfully) on the surrounding land and offer healthy leisure conditions to the working class e.g., [38]). However, the peripheral areas, as they do not ensure a certain return on investment, have so far remained outside the interest of most investors. In the context regarding the development of leisure industries, peripheral and green areas are increasingly valued and it can be assumed that the Green Deal can intensify these changes. The *ad hoc* use of extensively overgrown ruderal green areas is now possible (e.g., as recreational spaces) which, apart from promoting aesthetic non-obviousness, wilderness (as an effect of lack of public use) is encompassed by the aforementioned mechanism of profiting from decline. What can be seen as a novelty in the management of a city struggling with the problem of transformation (popularised in the last decade), which has an applied value for the inhabitants of Brzeszcze, is the use of the effects of disuse—ruin and succession—for building a local advantage by attracting users looking for "authenticity", "naturalness" and "greenness" (both residents and tourists). The combination of destructive market mechanisms and desirable expressions of social life takes an organised form—defined (by the minimum possible means) as publicly accessible spaces on post-industrial sites (Świątek [39] discusses the genesis of peripheral publicly accessible spaces as a result of changes to the urban structure and the forms that define them, mentioning, among others, obsolete spaces of industry). Additionally, as it was mentioned before, attracting users to a post-industrial area is a desirable phenomenon, not yet initiated and not yet estimated in the case of Brzeszcze, but probably possible to be brought about using small (relatively) financial outlays (As a reference example, the

project entitled Nature Enclave *Bobrowisko*—implemented in 2018 on the outskirts of Stary Sącz, within flooded gravel pit, which attracts around 200,000 visitors per year—is taken . The post-industrial succession site was adapted to the needs of visitors with a total investment of EUR 366,060 [40,41]).

*4.5. Minimal Interventions*

Minimal interventions are, in this context, a way to commit as little resources as possible to organise the improvement of the accessibility and the image of the site—they serve as a promotional tool (not to be confused with tools for sustainable change). Owing to them, the use of dilapidated buildings and abandoned land allows to make the most useful (and economically efficient) use of the uselessness of post-industrial "ruins"—difficult for traditional adaptation (in which value is combined with novelty and lack of exploitation traces). The significance of this initial stage of the project is twofold:

- firstly, it shows that a rational assessment of the condition of post-industrial facilities does not have to be tied to the idea of best possible use as being necessary to achieve in the initial stages of revitalisation;
- secondly, it points out that the search for solutions that take into account the need for (preliminary) adaptation of the post-industrial legacy, on the scale of local budgetary possibilities, helps to neutralise the conflict between those who advocate market mechanisms for the adaptation of heritage and those for whom the preservation of culture and identity is more important than material goods.

The essence of this phase of the project also emphasises the need for specific case study research as an important element for decision making, in the context of global changes which are not yet well visible at the local level. Last but not least, it promotes anticipatory decision making and the need to involve (also financially) local government as essential elements to have a potential impact on the future.

## 5. Conclusions

The challenge of achieving climate neutrality by 2050 entails economic transition in mining and energy regions. The social and environmental impacts of the energy transition on a European scale are reflected in the hundreds of thousands of people in need of support and in the countless areas in need of remediation, restoration and redevelopment. The climatic, environmental, social and economic challenges of the energy transition are also reflected at regional and local scales. The example of the mining town of Brzeszcze, which has been presented by the following paper, demonstrates that new challenges specific to the local level emerged at the stage of preliminary studies and work on preparing the site for revitalisation.

Before specifying the challenges, however, an analysis of EU policies showed that "green", i.e., ecology, biodiversity, quality crops—are important elements of transformation plans. A query of EU regulations, communications and strategies entitles us to conclude that nature conservation is supposed to ensure the "well-being" of the Europeans. However, compared to the material "well-being" (buildings and infrastructure) as well as immaterial "well-being" (privileges: vouchers, subsidies, incentives) provided by a "carbon-based" economy, this new "well-being" may be considered as less tangible and less material by those concerned.

In this context, the challenge is to find the ways of new redevelopment, compatible with the top-down policy, in a situation where the "green transition" faces understandable resistance of a significant group of inhabitants, professionally connected with mining. The application of minimal and targeted interventions enables the transformation of unvisited places into used ones, offers the chance to capitalise on the profits (social and probably economic) from the preservation of post-mining buildings—which are the material representation (in space) of the principles of a circular economy. As such, a quick and not

expensive action was accepted by the representatives of the local community as a challenge possible to undertake.

Considering the preliminary stage of the project, it must be stated that at this moment it is difficult to determine the effects and the possible degree of realisation regarding the assumptions of the socially just and "green" (environmentally improving) transformation of the city (it is also difficult to estimate the significance and the size of the critical situation caused by the closure of the mine in Brzeszcze and the neighbouring towns). The analysis including the outlining tendencies of the (perhaps unconscious) move of Brzeszcze towards the implementation of the "decline machine" indicates that the collapse of the mine can be used either to strengthen the position of the inhabitants of Brzeszcze as causal agents influencing the changes in their territory or to deprive them of their subjectivity. Preliminary research has proven that it is not too early to take anticipatory action, but it is also not (yet) too late. It is known, however, that the transformation of Brzeszcze already at the present time (VI 2021) requires the inhabitants to accept the challenge, which can be defined as the necessity to take anticipatory actions requiring a financial contribution.

In this context, one of the greatest challenges facing the expert team which undertook the work on the plan for transformation of Brzeszcze and revitalisation of the post-mining area can be defined. It is probably a risky attempt to propose new, material structures which can coexist with post-industrial sites and organise (enable the creation of) workplaces. Work, the nature of which will be in line with the principles of the Green Deal, circular economy and sustainable development.

Without reaching a consensus that risk is inherent in the transformation plan and that decaying post-industrial sites give the city of Brzeszcze an advantage in the Green Deal reforms (although they require action and public investment ahead of the market-driven "machinery of decline"), it would be difficult to find elements that sustain identity and unite the community—but for this, the greatest social challenge (at this stage of the project), the inhabitants of Brzeszcze proved ready. Another challenge is to maintain the sustainability of this position.

**Author Contributions:** Conceptualization, A.S.-Ś., A.O., M.C.; methodology, A.S.-Ś., A.O.; formal analysis, A.O., A.S.-Ś.; investigation, resources, A.S.-Ś., M.C., A.O.; writing—original draft preparation, A.S.-Ś., M.C., A.O.; writing—review and editing, M.C, A.O.; visualization, A.O.; supervision, M.C.; funding acquisition, M.C. All authors have read and agreed to the published version of the manuscript.

**Funding:** This research was funded by Initiative for Excellence—Research University in the frame of university grant entitled "Models of transition to a climate-neutral, circular economy for mining regions in transformation". Project contract number: 501.696.7995.

**Acknowledgments:** The authors would like to acknowledge Klaudia Zwolińska (AGH-UST) for her help in preparing the graphics.

**Conflicts of Interest:** The authors declare no conflict of interest. The funders had no role in the design of the study; in the collection, analyses, or interpretation of data; in the writing of the manuscript, or in the decision to publish the results.

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
