# Peer review of "Challenges of Coal Mining Regions and Municipalities in the Face of Energy Transition"

_energies, doi:10.3390/en14206674_

Round 1

Reviewer 1 Report

This article summarizes the problems facing the geographical areas of the EU affected by the closure of coal mines, with concrete reference to Polish communities, and the social, economic and environmental challenges posed by this transition. The paper presents very consistent information and makes scientifically justified proposals for the economic and social restructuring of the affected regions. The article is interesting, very well written and deserves to be disseminated through publication, as the case studies presented can be models of good practice and inspiration for other regions and communities affected by the closure of coal mines in the EU and the world.

Author Response

Dear Reviewer,

On behalf of the co-authors and myself, I sincerely thank you for the review!

best, anna

Reviewer 2 Report

The article is devoted to an actual topic. Due to the difficult environmental situation in the world, "green" solutions must increase. Most of the cities in which mining enterprises are located are monotowns, the economy and sociality of which is highly dependent on the operation of the enterprise. Resources are depleted. Therefore, mining enterprises finish their work. Also, many EC enterprises will finish their work in the near future due to environmental policy. The issues of economic and social stability in such cities are very difficult.

The following questions about this article:

 - the article does not provide an overview of the experience in other cities and countries where mining enterprises ceased to exist.

- it is not clear, what economic investments are required to implement the proposed solutions

- No.No. 21-25, 38, 39 in the references are not by English.

Author Response

Dear Reviewer,

On behalf of the co-authors and myself, I sincerely thank you for the review and the questions you asked. Answers are attached and below.

Comments and Suggestions for Authors

The following questions about this article:

  • the article does not provide an overview of the experience in other cities and countries where mining enterprises ceased to exist

We have in mind the examples of other cities, as well as mining regions, which have experienced the restructuring process (1990s) and are now experiencing the energy transition. Intentionally we left such a review and comparative analysis for the next stage and article, in which we will present the way to determine the optimal functions for the post-mining area in Brzeszcze. Mentioning only the cities that have experienced transformation will be too general information, while even a short description will significantly increase the volume of the already quite long article. Therefore, if you accept this, we will return to this topic in the next article, which will focus on redevelopment (revitalisation) of the post-mining area.

  • it is not clear, what economic investments are required to implement the proposed solutions

At this stage we cannot present details concerning economic investment, but in general we would like to “(…) propose new, material structures which can coexist with post-industrial sites and organise (enable the creation of) workplaces. Work, the nature of which will be in line with the principles of the Green Deal, circular economy and sustainable development.” – please see conclusion. However, as the main economic investment, on this preliminary stage of revitalization of Brzeszcze Coal Mine East area, is takeover of obsolete industrial building (even in not quite satisfying technical condition) by commune – to prevent demolition and starting “decline machine”.

  • No. 21-25, 38, 39 in the references are not by English.
  • Already translated into English.

In addition, we have changed the caption in Figure 5 by changing the word "municipality" to "commune."

Reviewer 3 Report

More than the original scientific article, it seems to me to be a popular article showing the problem of a region or city facing the problem of the transition from coal-based energy production to a green society. What I miss in the article is that there is no assessment of the impact of this change on GDP, I also see no statistical forecasts anywhere, of course an assessment of what an investment in green technology would bring. I think it would be very interesting to show the share of coal-based energy in gross domestic product. It would also be interesting to show the share of CO2 emissions in CO2 emissions. I am convinced that showing the share of CO2 emissions from coal and transport over the last few years would show an interesting picture. In my country, it turns out that emissions from transport are increasing, despite the introduction of strict standards, and emissions from coal remain the same. Perhaps in the field of coal mining, greater purity of emissions should be ensured, and this industry should be abandoned more slowly than in the EU. Maybe this would provide enough energy for the transition to electrification of traffic ... These answers, I would expect in an article with statistical representations.

Author Response

Dear Reviewer,

on behalf of the co-authors and myself, I want to thank you so much for the review – your questions and suggestions. Here are our answers:

What I miss in the article is that there is no assessment of the impact of this change on GDP, I also see no statistical forecasts anywhere, of course an assessment of what an investment in green technology would bring.

In the document "Energy and Climate Policy Scenario for Poland" (Analytical Appendix No. 2 to the "Poland’s National Energy and Climate Plan for years 2021-2030" [2019]), the macroeconomic impacts were assessed in the scenario including climate and energy targets, and additionally compared with the scenario without these targets. Answering the question about "assessment of the impact of this change on GDP" in the referenced document, it was found that the differences between the results of GDP levels in the two scenarios are not very large – GDP from 2030 is slightly higher in the scenario including climate-energy targets. Moreover, for three decades (1988-2018), it is observed that the increasing value of GDP does not entail an increase in the level of greenhouse gas emissions. Between 1988 and 2018, there was a 32% reduction in GHG emissions, with a nearly 3-fold increase in GDP. (Please see changes in Chapter 1, line 95).

In the context of the question asked in the Introduction, concerning whether green jobs will compensate for those lost in the mining and energy industry, we quoted the forecast presented in the SCENARIO OF ENERGY AND CLIMATE POLICY, noting, however, that these are estimates. We know unofficially that redundancies from mining and energy companies are not compensated in the RES sector, offered by the same companies. It is also too early to assess the situation comprehensively on a national scale. (Please see changes in Chapter 1, lines 45-49).

I think it would be very interesting to show the share of coal-based energy in gross domestic product.

Unfortunately, the information about the share of coal-based energy in GDP is not available. In statistics, however, we could find information that the sectors of generation and supply of electricity, gas, steam and hot water and mining and quarrying create about 4.5% of gross value added of Polish GDP.

It would also be interesting to show the share of CO2 emissions in CO2 emissions. I am convinced that showing the share of CO2 emissions from coal and transport over the last few years would show an interesting picture.

Yes, you are right! The analysis of historical data shows increasing CO2 emissions from transport and decreasing CO2 emissions from industry (including mining and energy industry). In the context of the discussion in this article about preserving or decommissioning mine infrastructure, data on the impact of construction sector on CO2 emissions are relevant and is underlined in the table. This has to do not only with decommissioning of mines, but investing in modern technologies of coal processing and combustion. We have supplemented the article with historical data and CO2 emission forecasts. The forecasts take into account the implementation of the scenario of the Polish Energy and Climate Policy, which assumes, among other aspects, a decrease in emissions. (Please see changes in Chapter 1, lines 89-95 and Table 1).

In addition, we have changed the caption in Figure 5 by changing the word "municipality" to "commune."

Reviewer 4 Report

This paper addresses the important topic of social and environmental justice in coal mining areas during the low-carbon energy transition. The authors report on field studies in a coal mining area of Poland and highlight a number of challenges and opportunities facing the local population. They also place the study in the context of the EU's Green Plan.

The main weakness of the paper is its structure.

The Introduction should more clearly state the specific aims of the study

The key concepts to be applied (beyond the Green Plan)  are not presented clearly and sufficiently early in the paper in a way that provides and analytical framework for the analysis. For example: 

- Section 4.1 sets out some important principles and states the aim of the paper. This does not fit under the heading 'results'. Surely the text of this section should be much earlier in the paper.

  • In contrast section 4.2 does address 'results', however it includes concepts such as 'growth machine' that have not been introduced before. Again, these basic  concepts and the supporting literature should be presented earlier in the paper.

I suggest that Section 2 be used to present all the concepts to be used in the paper, with supporting literature, in addition to the Green Plan.  This can then be used as the basis for an analytical framework. All subsequent text would then be structured around the elements of this framework so that each element/topic is addressed separately.

Minor comment:

- some paragraphs are too long, e.g. page 11

Author Response

Dear Reviewer,

On behalf of the co-authors and myself, I want to thank you so much for the review!

We have changed the structure of the paper due to reviewer suggestions. We have underlined aims of the study in introduction. (Please see changes in Chapter 1, lines 140-143).

In section 4.1. we have left reminding note about aims. Discussion in this section (4.1), has been planned as preliminary research to emphasize how one of flagship ideas of Green Deal (ideas of greenery) is being understood in coal-mine town of Brzeszcze – what kinds of benefits can be seen by interlocutors (self-managing, recreational, climate-friendly) and what threats are being stressed (“museification"). We added clarification notes to this paragraph.

Concepts of “growth machine” and “decline machine” have been cut from paragraph 4.2 and placed in point 2 of paper, as it was suggested. (Please see changes in Chapter 2, lines 249-260).

Due to these changes in structure paragraph on page 11 was shortened.

In addition, we have changed the caption in Figure 5 by changing the word "municipality" to "commune."

Round 2

Reviewer 3 Report

The article is now suitable for publication!

Reviewer 4 Report

the reviewers comments have been addressed and the paper may now be published